# Markers of protein-energy wasting and physical performance in haemodialysis patients: A cross-sectional study

**Karsten Vanden Wyngaert**[1], **Bert Celie**[1], **Patrick Calders**[1], **Sunny Eloot**[2], **Els Holvoet**[2], **Wim Van Biesen**[2], **Amaryllis H. Van Craenenbroeck**[3,4]*

1 Department of Rehabilitation Sciences, Faculty of Medicine and Health Sciences, Ghent University, Ghent, Belgium, 2 Department of Internal Medicine, Renal Division, Ghent University Hospital, Ghent, Belgium, 3 Laboratory of Experimental Medicine and Paediatrics, University of Antwerp, Antwerp, Belgium, 4 Department of Nephrology and Renal Transplantation, University Hospitals Leuven, Leuven, Belgium

* amaryllis.vancraenenbroeck@uzleuven.be

**Data Availability Statement:** All relevant data are within the paper and its Supporting Information files.

## Abstract

### Background

Physical impairments are common in uraemia, as reflected by the high risk of falls of haemodialysis (HD) patients. Furthermore, these patients often suffer from malnutrition.

### Objective

Up to now, it is unknown which aspects of physical performance are predominantly driven by malnutrition in HD patients. As this answer could steer different interventions, the aim of this study was to evaluate the cross-sectional relationship between nutritional status, muscle strength, exercise capacity and the risk of falls.

### Methods

This study recruited HD patients between December 2016 and March 2018 from two hospital-based and five satellite dialysis units (registration number on clinicaltrial.gov: NCT03910426). The mini-nutritional assessment scale as well as objective measures of protein-energy wasting were obtained (total iron-binding capacity, total protein levels, and CRP). Physical assessment included muscle strength (quadriceps, handgrip force, and sit-to-stand test), exercise capacity (six-minute walking test) and the risk of falls (Tinetti, FIC-SIT, and dialysis fall index). Their interrelationship was analysed by ridge regression models.

### Results

Out of 113 HD patients (mean age 67 years ± 16.1, 57.5% male) 36.3% were malnourished according to the mini-nutritional assessment scale and a majority had impaired quadriceps force (86.7%), six-minute walking test (92%), and an increased risk of falls (73.5%). Total protein and CRP levels were identified as relevant nutritional factors in the association with physical performance. Nutritional parameters explained 9.2% of the variance in the risk of

**Funding:** The author(s) received no specific funding for this work.

**Competing interests:** The authors have declared that no competing interests exist.

falls and 7.6% of the variance in exercise capacity. No conclusive association was found between nutritional status and muscle strength.

## Conclusion

Protein-energy wasting is a determinant of *the risk of falls* and *exercise capacity* in patients on HD. The association between malnutrition and *muscle weakness* remains inconclusive.

## Introduction

Physical impairments and haemodialysis (HD) therapy itself are major barriers for physical activity in patients with end-stage kidney disease (ESKD) [1, 2]. Physical inactivity leads to a high risk for cardiovascular (CV) disease and results in a more extensive deterioration of physical performance [3]. Although the manifestation of physical impairments (i.e. muscle weakness, exercise intolerance and an increased risk of falls) in ESKD is heterogeneous, the downward spiral of physical inactivity can be positively influenced by a physical rehabilitation programme [4, 5], whether or not supplemented with nutritional interventions [6, 7]. Unravelling the role of the different players is important as a better understanding of the underlying mechanisms might lead to better exercise training outcomes in patients on HD.

According to the International Society of Renal Nutrition and Metabolism (ISRNM), protein-energy wasting (PEW) is defined as a status of nutritional and metabolic impairments characterized by loss of systemic body protein and energy stores, resulting in a decrease in muscle and fat mass [8]. In uraemia, strong dietary restrictions and low protein intake substantially contribute to this phenotype [9–11]. Although a decrease in muscle mass directly results in muscle weakness in healthy subjects, only a fraction of the variance in muscle strength is explained by muscle mass in patients with ESKD [12]. Moreover, muscle strength rather than muscle mass seems to be affected by the presence of PEW [13]. Additionally, PEW along with impairments in physical performance contribute to a remarkably high prevalence of frailty in patients with ESKD [14]. Nevertheless, to date, the relationship between malnutrition and the different domains of physical performance remains poorly understood [15].

The aim of this study is to examine the cross-sectional relationship between markers and scoring systems for nutritional status and different measures of physical performance in prevalent HD patients. Our hypothesis is that PEW is more closely related to measures that reflect broad daily physical performance (e.g. the risk of falls) compared to analytical measures of physical performance (e.g. quadriceps peak torque), albeit based on the broad impact of PEW.

## Materials & methods

Consecutive patients on maintenance HD in two dialysis centres (including two hospital-based and five satellite dialysis units) were screened for eligibility between December 2016 and March 2018. Exclusion criteria were age < 18 years, pregnancy, inadequate motor and verbal responses to verbal commands and questions, and recent (< 6 months) surgical musculoskeletal interventions that could bias physical tests. Patients with physical inabilities (e.g. wheelchair bound or amputations) were given the worst possible score for the tests they failed to complete.

This study is part of a larger study examining the determinants of quality of life and mortality in patients with ESKD (registration number on clinicaltrial.gov: NCT03910426). The study complies with the Declaration of Helsinki, was approved by the local ethics committees

(project number Ghent EC B670201525559; 15-OCT-2015 and Antwerp EC B300201422642; 07-DEC-2016), and written informed consent was obtained from all participants.

## Anthropometric measures and characteristics

Baseline clinical data and anthropometric measures were obtained from the electronic medical records and the Davies comorbidity score was calculated [16]. Blood pressure was evaluated with a single measurement at the opposite upper arm to the side of the vascular access before the dialysis session prior to the physical performance assessment.

## Nutritional status

The use of multiple nutritional markers has been recommended in assessing nutritional status [17]. Nutritional status was quantified both subjectively and objectively using the Mini Nutritional Assessment (MNA) Long Form [18, 19] on the one hand and body mass index (BMI), C-reactive protein (CRP) and biochemical indicators such as total iron binding capacity (TIBC) and serum total protein on the other [20–22]. The MNA was obtained by a study nurse and included 2 sections: a screening section (MNA-short form, /14) which was complemented by a more profound assessment section (/16), resulting in a global score on 30 (MNA-long form). The *screening* section addresses food intake, weight loss, mobility, neuropsychological problems, BMI, and health status over the last three months, whereas the *in-depth assessment* section comprises questions related to the place of residence, number of prescribed medications, skin ulcers, eating and drinking behaviour, the subjective appreciation of nutritional status, and mid-arm and calf circumference. To increase statistical power, patients were classified as malnourished or at risk for malnutrition based on the median of the MNA-score of patients scoring ≤ 23.5. Accordingly, patients were identified with malnutrition (≤ 19.5), malnutrition risk (20–23.5) and normal nutritional status (≥ 24) [18]. The term PEW is used when an association included the MNA and at least one objective measure associated with loss of body protein and fuel reserves (e.g. CRP, serum total protein and TIBC) [23].

## Physical assessments

All individual physical assessments were done within two days. The sequence of assessments was randomized using opaque envelopes. Muscle strength evaluation was performed before the dialysis session, while exercise capacity and the risk of falls were assessed either before dialysis or on non-dialysis days (in patients' home setting). A minimum 3-minute pause between tests was respected.

**The risk of falls.** For assessment of the risk of falls, a combination of physical testing, scoring lists and demographic data was used in a slightly adapted version of the Dialysis Fall Risk Index (DFRI, see S1 Table) [24]. The physical examinations included the following: (1) The Frailty and Injuries Cooperative Studies of Intervention Technique (*FICSIT*) was used to examine static balance (time) based on seven positional challenges; i.e. eyes open and closed with feet closely together, semi-tandem and full tandem stand and standing on the dominant leg with eyes open [25]; (2) The *Tinetti* test is considered the gold standard for examining gait dysfunctions based on 7 items: the initiation of gait, step length and height, step symmetry, step continuity, distinguished path, trunk and walking stance [26]. Patients scoring < 11 on 12 on the Tinetti test are considered to be at a higher risk of falls [27]; (3) For the five repetition *Sit-to-Stand test* (STS), patients were instructed to get from a seated to a standing position for 5 times as rapidly as possible with their arms folded across the chest [28]. A cut-off value of 15 seconds is associated with an increased risk of falls [29] and patients unable to perform the test were scored as > 50 seconds. With regard to the original *DFRI*, the following adaptations were

made: (1) a 2.9 mg/dl instead of 1.0 mg/dl cut-off point for CRP; (2) MNA indicator scores were used as an alternative for the Geriatric Nutritional Risk Index [30, 31]; (3) six-minute walking test (6MWT) replaced the '4 meter time to walk' test and (4) the 'inquiry about fall' section was replaced by the Tinetti test [32]. The mean arterial pressure was calculated by diastolic blood pressure + 1/3(systolic blood pressure—diastolic blood pressure).

**Muscle strength.** A handheld dynamometer (Microfet; Biometrics, Almere, the Netherlands) was used to evaluate *quadriceps* isometric peak torque during 5 seconds in a seated position with knees and hips 90˚ flexed (intraclass correlation coefficient (ICC) of 0.94 [33, 34]. Manual resistance with fixation of the dynamometer to the anterior tibia of the dominant leg just proximal to the malleoli was applied. *Handgrip* force was measured using a JAMAR Hydraulic Hand Dynamometer according to the American Society of Hand Therapists guidelines (ICC ≥ 0.93) [35]. Patients were seated with their elbow 90˚ flexed next to their body, wrist in neutral position and were asked to perform a maximal isometric contraction for 5 seconds [36]. The contralateral arm with regard to the vascular access was used.

Both quadriceps and handgrip force were carried out in triplicate; the best result was expressed as absolute value and as percentage of the predicted value based on age and gender [33, 37]. The lower limit of normal for the quadriceps and handgrip force was set on 80% of the predicted value.

**Exercise capacity.** The 6MWT was performed following the American Thoracic Society guidelines (ICC = 0.90) [38]. Patients were instructed to walk as fast as possible for 6 minutes, walking aids were allowed and recorded. Results were expressed as absolute value and as percentage to the predicted value, based on Duncan's equation for adults aged between 50–85 years [39]. The lower limit of normal was set on 80% of the predicted value. Moreover, a 300m cut-off point was used for dichotomisation, as this indicates a worse prognosis and higher mortality in populations comparable to ESKD [40, 41]. Patients unable to perform the tests were scored as 0 meters.

## Statistical analysis

IBM Statistical Package for the Social Sciences version 24 (SPSS 24) and R were used for the statistical analyses. Variables are reported as mean ± standard deviation (SD), median and interquartile range [25th; 75th percentage] or as number and percentage when appropriate. Data between groups were compared by using univariate analysis of variance or the Kruskal-Wallis as the nonparametric equivalent. *Post hoc* comparisons were corrected using the Scheffe's or pairwise comparison (Mann-Whitney U) test for parametric and nonparametric data respectively. A ridge regression method was used to examine the association between nutritional status and measures of muscle strength, exercise capacity and the risk of falls. By using L2 (ridge) regularization techniques, univariate associations can be examined between a dependent variable (e.g. a measure of physical performance) and a matrix of potentially collinear independent variables (i.e. nutritional status by subjective and objective measures) without overestimating the association. The collinearity penalty allows us to find independent associations that indicate the involvement of different aspects of malnutrition in impaired physical performance. The ridge regression was performed using the lmridge package in R and the general cross-validation method was used to estimate the optimal k [42]. Nutritional measures included in the DFRI (i.e. MNA and CRP) were excluded in the analysis of this index. The spearman rank correlation coefficient was used in correlative analysis. Patients with missing data on primary and secondary outcome measures were excluded from the final analysis.

## Results

### Demographics

122 patients were enrolled in this study and 9 were excluded based on missing data on questionnaires (n = 6) and on measures of physical performance (n = 3). Accordingly, 113 patients were included in this study. Patient characteristics and outcome measures did not differ between the excluded patients and the study cohort. The study population (age 67 ± 16.1 years, 57.5% male and a range of dialysis vintage between 1 and 191 months) was representative for a cohort of HD patients (Table 1). Impaired quadriceps force was prevalent in the total cohort and the average score was 53.8% of the predicted value. In line, muscle performance of the lower limbs (assessed by STS) was decreased, with 61% of patients scoring above the upper limit of normal and indicating an increased risk of falls. Muscle weakness was less pronounced in the upper limbs. In general, this cohort had an increased risk of falls (by DFRI, 73.5%) and exercise intolerance (by 6MWT, 68.1%). Indeed, the majority of patients were classified as having a bad prognosis based on a physical surrogate measure (6MWT < 300m). Furthermore, these patients were older, more likely to be female, had lower BMI, higher CRP, a higher number of prescribed medications, and scored worse on all other domains of physical performance (Table 1). No differences in measures of physical function were found between patients with and without diabetes (S2 Table).

Only a minority of participants was rated as well-nourished, whereas 47.8% and 36.3% were identified as being at risk for malnutrition and malnourished respectively. Of note, MNA scores did not significantly differ between patients with a good and a bad prognosis based on the 6MWT.

### Determinants of physical performance to malnutrition

Table 2 shows the characteristics of the study population according to nutritional status based on MNA. No differences were found for muscle strength between the three groups of nutritional status. Patients identified with normal nutritional well-being noted a lower risk of falls compared to those without. Also, malnourished participants had lower exercise capacity than expected for their age and gender compared to the participants at risk for malnutrition, as depicted in Fig 1.

A ridge regression analysis of the measures of nutritional status to the different domains of physical performance is demonstrated in Table 3. Nutritional status explained 9.2% of the variance in the risk of falls, as assessed by Tinetti. No associations were found with muscle strength and exercise capacity. However, a tendency towards an association was observed between nutritional status and exercise capacity, as assessed by 6MWT ($R^2 = 0.05$). A more detailed analysis showed the involvement of total protein levels and CRP in the association with the risk of falls and exercise capacity respectively (see S3–S5 Tables). Note that the association with the risk of falls was independent of age, gender and level of comorbidity.

Muscle strength ($R^2 = 0.07$) and exercise capacity ($R^2 = 0.08$) as expected for the patients' age and gender were associated with nutritional status (Table 4). Remarkably, apart from MNA, BMI and CRP contributed to these associations respectively (see S6 and S7 Tables).

## Discussion

The present study explored the impact of nutritional status on physical performance in prevalent HD patients (67.5 years ± 16.1). As expected, the prevalence of impaired nutritional status and physical impairments in the studied population was strikingly high. PEW, represented by total protein levels and MNA, explained 9.2% of the variance in *gait quality* as assessed by

**Table 1. Patient characteristics according to prognosis based on a physical surrogate.**

| Variable | Total (n = 113) | Good prognosis (6MWT > 300m, n = 46) | Bad prognosis (6MWT < 300m, n = 67) | p value |
|---|---|---|---|---|
| **Demographics** | | | | |
| Age (years) | 67.5 ± 16.1 | 58.2 ± 18.0 | 73.8 ± 10.9 | <0.001 |
| Sex (male) | 65 (57.5) | 32 (69.6) | 33 (49.3) | 0.003 |
| BMI (kg/m²) | 26.1 ± 5.4 | 25.2 ± 4.7 | 26.7 ± 5.8 | 0.135 |
| BMI <18 | 3 (2.7) | 1 (2.2) | 2 (3.0) | |
| BMI 18–24 | 51 (45.1) | 25 (54.3) | 26 (38.8) | |
| BMI 25–29 | 35 (31.0) | 14 (30.4) | 21 (31.3) | |
| BMI ≥30 | 23 (20.5) | 6 (13.1) | 18 (26.9) | |
| DBP (mmHg) | 65.4 ± 16.0 | 71.2 ± 18.5 | 62.8 ± 14.1 | 0.023 |
| SBP (mmHg) | 138.2 ± 21.6 | 142.1 ± 21.1 | 136.5 ± 21.8 | 0.208 |
| ΔMAP (mmHg) | 0.17 ± 15.5 | -0.12 ± 15.5 | 0.36 ± 15.6 | 0.873 |
| MAP pre-dialysis (mmHg) | 89.7 ± 15.3 | 92.0 ± 16.7 | 88.2 ± 14.2 | 0.223 |
| MAP post-dialysis (mmHg) | 89.3 ± 17.8 | 90.8 ± 17.0 | 88.3 ± 18.3 | 0.474 |
| Dialysis vintage (months) | 22.5 [10.3; 49.8] | 20.5 [9.8; 35.3] | 25.0 [10.8; 64.3] | 0.169 |
| Dialysis vintage <24 | 57 (50.5) | 26 (56.5) | 32 (47.8) | |
| Dialysis vintage 24–47 | 26 (23.0) | 12 (26.1) | 13 (19.4) | |
| Dialysis vintage ≥48 | 30 (26.5) | 8 (17.4) | 22 (32.8) | |
| Number of prescribed medications (n) | 13.6 ± 3.7 | 12.6 ± 3.9 | 14.3 ± 3.5 | 0.016 |
| Aetiology of CKD (n) | | | | 0.062 |
| Diabetic nephropathy | 30 (26.5) | 6 (13.0) | 24 (35.8) | |
| Hypertension or angiosclerosis | 40 (35.4) | 16 (34.8) | 24 (35.8) | |
| ADPKD | 6 (5.4) | 4 (8.7) | 2 (3.0) | |
| Other | 37 (32.7) | 20 (43.5) | 17 (25.4) | |
| Davies comorbidity score (0–7) | 2 [1; 3] | 2 [2; 3] | 1 [0; 2] | <0.001 |
| Ethnicity | | | | 0.456 |
| Black | 3 (2.7) | 3 (6.5) | 0 (0.0) | |
| White | 110 (97.3) | 43 (93.5) | 67 (100.0) | |
| Other | 0 (0.0) | 0 (0.0) | 0 (0.0) | |
| **Nutritional and physical assessments** | | | | |
| Quadriceps force (N) | 180 ± 75 | 222.6 ± 78.8 | 136.7 ± 65.1 | <0.001 |
| Relative value (% to predicted) | 53.8 ± 17.8 | 56.4 ± 18.0 | 48.4 ± 22.0 | 0.035 |
| Patients with pathological value (n) | 98 (86.7) | 40 (87.0) | 58 (86.6) | 0.895 |
| Handgrip force (kg) | 28.8 ± 11.1 | 36.1 ± 10.0 | 23.7 ± 8.8 | <0.001 |
| Relative value (% to predicted) | 91.7 ± 30.7 | 94.7 ± 20.6 | 90.6 ± 36.1 | 0.439 |
| Patients with pathological value (n) | 39 (34.5) | 9 (19.6) | 30 (44.8) | 0.060 |
| DFRI (/12) | 5.9 ± 3.0 | 3.1 ± 2.2 | 7.8 ± 1.8 | <0.001 |
| Patients at increased risk of falls (n) | 83 (73.5) | 17 (37.0) | 66 (98.5) | <0.001 |
| Tinetti (/12) | 11.0 [5.5; 12.0] | 12.0 [12.0; 12.0] | 7.0 [0.0; 10.0] | <0.001 |
| Patients at increased risk of falls (n) | 55 (48.7) | 2 (4.3) | 53 (79.1) | <0.001 |
| Sit-to-Stand (s) | 23.0 [12.0; 50.0] | 12.0 [10.0; 15.3] | 50.0 [23.0; 50.0] | <0.001 |
| Patients at increased risk of falls (n) | 78 (69.0) | 14 (30.4) | 64 (95.5) | <0.001 |
| FICSIT | 15.0 [8.0; 21.0] | 22.0 [16.0; 26.0] | 10.0 [2.0; 15.0] | <0.001 |
| 6MWT (meters) | 236 [66.5; 396.5] | 455 [400.0; 514.8] | 130 [0.0; 239.0] | <0.001 |
| Relative value (% to predicted) | 44.1 [12.7; 60.3] | 67.8 [59.2; 79.7] | 24.4 [0.0; 46.2] | <0.001 |
| Patients with pathological value (n) | 104 (92.0) | 37 (80.4) | 67 (100.0) | <0.001 |
| Patients scoring <300m (n) | 77 (68.1) | 0 (0.0) | 67 (100.0) | |

(*Continued*)

**Table 1.** (Continued)

| Variable | Total (n = 113) | Good prognosis (6MWT > 300m, n = 46) | Bad prognosis (6MWT < 300m, n = 67) | p value |
|---|---|---|---|---|
| Mini-nutritional assessment | 20.7 ± 2.9 | 21.1 ± 3.1 | 20.4 ± 2.7 | 0.262 |
| Normal nutritional status | 18 (15.9) | 9 (19.6) | 9 (13.4) | |
| At risk of malnutrition | 54 (47.8) | 21 (45.7) | 33 (49.3) | |
| Malnourished | 41 (36.3) | 16 (34.8) | 25 (37.3) | |
| C-reactive protein (mg/L) | 4.3 [2.7; 10.0] | 2.9 [1.4; 6.7] | 5.7 [2.9; 10.8] | 0.003 |
| Total iron-binding capacity (µg/dL) | 240.4 ± 76.6 | 238.8 ± 77.6 | 241.5 ± 76.6 | 0.859 |
| Serum total protein (g/L) | 65.2 ± 6.1 | 64.7 ± 5.4 | 65.5 ± 6.6 | 0.508 |

Data are reported as mean ± standard deviation, median [25%; 75%] or as number (percentage) as appropriate; patients were allocated to a good or poor functional prognosis groups based on 6MWT; p-values from ANOVA were reported for normal distributed parameters, otherwise they were reported from the Kruskal-Wallis test. Abbreviations: ADPKD, autosomal dominant polycystic kidney disease; BMI, body mass index; CVD, cardiovascular disease; DBP, diastolic blood pressure; DFRI, dialysis fall risk index; SBP, systolic blood pressure; Δ, difference pre- to post-dialytic blood pressure.

Tinetti. Measures of inflammatory as well as nutritional status were associated with *functional exercise capacity* and *prognosis* based on a surrogate measure. Finally, although a relationship between impaired nutritional status and *muscle weakness* could not be confirmed, BMI was

**Table 2. Patient characteristics according to the nutritional status.**

| Variable | Normal nutritional status (n = 18) | Impaired nutritional status (n = 95) | p | Impaired nutritional status | | p (3 groups) |
|---|---|---|---|---|---|---|
| | | | | At risk of malnutrition (n = 54) | malnutrition (n = 41) | |
| Age (years) | 70.2 ± 10.3 | 67.0 ± 17.0 | 0.93 | 69.9 ± 13.7 | 63.1 ± 20.0 | 0.351 |
| Male sex (%) | 11 (61.1) | 54 (57.0) | 0.74 | 32 (59.3) | 22 (53.7) | 0.814 |
| BMI (kg/m$^2$) | 30.1 ± 4.6 | 25.3 ± 5.2 | <0.001 | 27.2 ± 4.9 | 22.9 ± 4.6 | <0.001 [a] |
| CRP (mg/L) | 3.0 [2.9; 8.8] | 4.7 [2.6; 10.5] | 0.39 | 5.4 [2.4; 11.0] | 2.7 [2.9; 9.2] | 0.569 |
| TIBC (µg/dL) | 215.6 ± 65.0 | 245.2 ± 78.1 | 0.10 | 251.8 ± 79.8 | 236.4 ± 75.9 | 0.209 |
| Total protein (g/L) | 65.0 ± 5.9 | 65.3 ± 6.2 | 0.86 | 67.1 ± 5.3 | 62.7 ± 6.5 | 0.002 [b] |
| Dialysis vintage (months) | 31.5 [10.5; 71.0] | 21.0 [10.0; 49.0] | 0.26 | 20.0 [10.5; 49;5] | 23.0 [9.0; 42.5] | 0.488 |
| Number of prescribed medications (n) | 12.1 ± 4.0 | 13.9 ± 3.6 | 0.09 | 13.5 ± 3.9 | 14.3 ± 3.1 | 0.118 |
| Davies comorbidity score (0–7) | 2 [1; 3] | 2 [1; 3] | 0.89 | 2 [1; 3] | 2 [0.5; 3] | 0.529 |
| Quadriceps force (N) | 183.9 ± 81.5 | 178.7 ± 74.4 | 0.88 | 193.6 ± 69.3 | 157.4 ± 77.1 | 0.060 |
| Quadriceps force (%) | 50.2 ± 20.7 | 54.5 ± 17.2 | 0.35 | 57.8 ± 15.1 | 49.6 ± 19.0 | 0.063 |
| Handgrip force (kg) | 32.3 ± 9.6 | 28.1 ± 11.3 | 0.08 | 29.7 ± 10.8 | 26.0 ± 11.6 | 0.076 |
| Handgrip force (%) | 97.4 ± 25.7 | 90.6 ± 31.6 | 0.19 | 93.8 ± 28.4 | 86.4 ± 35.2 | 0.163 |
| DFRI (/12) | 4.6 ± 3.0 | 6.1 ± 3.0 | 0.042 | 6.1 ± 2.7 | 6.2 ± 3.4 | 0.099 |
| Tinetti (/12) | 11.0 [8.5; 12.0] | 10.0 [5.0; 12.0] | 0.20 | 10.0 [5.8; 12.0] | 10.0 [2.0; 12.0] | 0.455 |
| FICSIT (/28) | 17 [14; 23] | 15 [6; 21] | 0.11 | 15 [8; 20] | 14 [3; 22] | 0.258 |
| Sit-to-Stand (s) | 17 [11; 29] | 30 [12; 50] | 0.08 | 23 [12; 50] | 50 [12; 50] | 0.116 |
| 6MWT (m) | 290 [201; 367] | 220 [0; 400] | 0.23 | 239 [98; 405] | 144 [0; 389] | 0.249 |
| 6MWT (%) | 49.3 [40.8; 70.7] | 41.8 [0.0; 59.3] | 0.05 | 44.6 [19.1; 66.3] | 29.0 [0.0; 55.7] | 0.021 [b] |

Data are reported as mean ± standard deviation; p-values from ANOVA were reported for normal distributed parameters, otherwise they were reported from the Kruskal-Wallis test. Abbreviations: 6MWT, six-minute walking test; BMI, body mass index; DFRI, dialysis fall risk index; CRP, C-reactive protein; TIBC, total iron binding capacity.

[a] p<0.05 patients without or at risk for malnutrition vs. malnourished patients.

[b] p<0.05 at risk for malnutrition vs. malnourished patients.

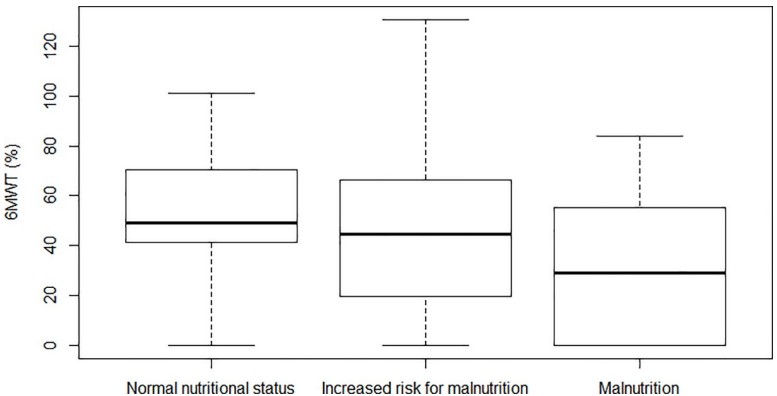

**Fig 1. Exercise capacity between groups of nutritional status.** Boxplots of the relative 6MWT for patients identified with normal nutritional status (MNA $\geq$ 24), at risk for malnutrition (MNA 20–23.5) and with malnutrition (MNA $\leq$ 19.5).

identified as an important contributor to quadriceps strength. Despite comparable muscle strength, patients with impaired nutritional status had lower exercise capacity than well-nourished patients.

Whereas various definitions of PEW are available in scientific literature, they all disclose the extraction from energy stored in proteins, such as in the muscles, as compensation for an insufficient energy intake [43]. Consequently, this whole body protein imbalance results in a reduced turnover of contractile proteins, muscle mass and force generating capacity, leading to physical impairments such as muscle weakness [44, 45]. In this study, the MNA long form is used which is a reliable tool in the screening, differentiation and diagnosis of PEW in patients with ESKD [46, 47]; moreover, in line with recent recommendations, objective measures of nutritional status are quantified as well [48]. Different measures that derive from different physiological systems enable us to differentiate between the pathways of malnutrition and PEW to physical impairments; such as markers of protein balance and iron homeostasis that

**Table 3. Association between measures of nutritional status and physical performance.**

| Outcome | Model 1 | | Model 2 (Model 1 + Age + Gender) | | Model 3 (Model 2 + Comorbidity index) | |
|---|---|---|---|---|---|---|
| | $R^2$ | *p* value | $R^2$ | *p* value | $R^2$ | *p* value |
| Quadriceps strength (N) | 0.047 | 0.145 | 0.374 | < 0.001[a] | 0.394 | < 0.001[a] |
| Handgrip strength (kg) | 0.018 | 0.311 | 0.526 | < 0.001[a] | 0.544 | < 0.001[a] |
| Tinetti (/12) | 0.092 | 0.011[b] | 0.190 | < 0.001[b] | 0.196 | < 0.001[b] |
| FICSIT (/28) | 0.033 | 0.169 | 0.241 | < 0.001[a] | 0.245 | < 0.001[a] |
| Sit-to-Stand (s) | 0.061 | 0.046[a] | 0.254 | < 0.001[a] | 0.279 | < 0.001[a] |
| DFRI (/12) | 0.010 | 0.301 | 0.034 | < 0.118 | 0.231 | < 0.001 |
| 6MWT (m) | 0.050 | 0.079[b] | 0.325 | < 0.001[a] | 0.354 | < 0.001[a] |

The following variables were introduced in ridge regression model 1: mini-nutritional assessment scale, total protein levels, total iron-binding capacity, C-reactive protein and body mass index; age and sex were added to model 1, resulting in Model 2

[a] $\geq$ 1 measure of nutritional status contributed to the overall effect size

[b] $\geq$ 1 objective measure of protein-energy wasting contributed to the overall effect size, albeit in addition to the mini-nutritional assessment scale (p < 0.05). Abbreviations: 6MWT, six-minute walking test; DFRI: dialysis fall risk index.

**Table 4. Association between measures of nutritional status and physical performance as expected for the patients' age, gender and height.**

| Outcome | Model 1 | | Model 2 | |
|---|---|---|---|---|
| | $R^2$ | *p* value | $R^2$ | *p* value |
| Quadriceps strength (%) | 0.066 | 0.034[b] | 0.068 | 0.044[b] |
| Handgrip strength (%) | 0.009 | 0.487 | 0.009 | 0.540 |
| 6MWT (%) | 0.076 | 0.024[b] | 0.130 | 0.003[a] |

The following variables were introduced in the ridge regression model 1: mini-nutritional assessment scale, total protein levels, total iron-binding capacity, C-reactive protein and body mass index; The davies comorbidity score was added to model 1 in model 2

[a] $\geq$ 1 measure of nutritional status contributed to the overall effect size

[b] $\geq$ 1 objective measure of protein-energy wasting contributed to the overall effect size, albeit in addition to the mini-nutritional assessment scale (p < 0.05). Abbreviations: 6MWT, six-minute walking test.

are associated with frailty and anaemia, respectively, and markers of inflammation which are involved in the malnutrition-inflammation-atherosclerosis (MIA) syndrome [48–50].

Serum protein levels are prognostic factors related to frailty in patients with ESKD [50]. Moreover, a negative protein balance as well as malnutrition contribute to a frailty prevalence of > 60% in patients with ESKD [51, 52]. Frailty is defined as a decline in physical resilience to stressors, resulting in a substantially decreased ability to cope with illness and in general health deterioration. Consequently, frailty is associated with an increased risk of adverse outcomes such as unintentional weight loss, functional degradation, delirium, slow walking speed, and an increased risk of falls [53–55]. In line with this definition, the present study reports an association of PEW (by MNA and total protein levels) with the risk of falls in ESKD patients. Hence, PEW and frailty may promote an already increased risk of falls, which will contribute to a high risk for hospitalization, disability and death in patients on HD [56, 57]. Interestingly, an increased risk of falls as well as malnutrition are well-established treatable aspects of frailty in patients with chronic kidney disease [52]. Furthermore, the multidisciplinary management of frailty, including exercise and nutritional interventions is effective in frail elderly adults and is recommended by the European Renal Best Practice working group [58, 59]. However, to our knowledge, it has not yet been examined whether fall prevention can be improved by improving nutritional status in patients with ESKD.

Persistent low-grade inflammation is a well-established component in the ESKD phenotype for PEW and CV disease [60]. In particular, an increase in pro-inflammatory and catabolic agents due to the upregulation of the immune system results in a high energy consumption and negative protein balance [61, 62]. Additionally, these inflammatory markers indicate endothelial dysfunction and vascular remodelling, resulting in a high risk for CV disease [63]. As a consequence to the convergent input of inflammation in the pathophysiology of PEW and CV disease, the term malnutrition-inflammation-atherosclerosis syndrome was established in patients with ESKD [64]. Remarkably, a prevalence of MIA of 53.9% is reported in patients on HD [65]. Consistent with CV impairments being the cornerstone of MIA, the finding that PEW (by MNA and CRP) associates with impairments in cardiorespiratory function, assessed via the 6MWT, is not unexpected. Furthermore, a correlation between inflammation and exercise capacity but not muscle strength, suggests that inflammation will affect physical performance by other mechanisms, for example endothelial dysfunction, rather than muscle wasting [63, 66]. Notwithstanding MIA, inflammation is associated with risk factors for falls in HD patients as well, such as frailty, autonomic and peripheral neuropathy, and hypotensive episodes [24, 67].

Interestingly, despite differences in age, BMI and CRP levels between patients with and without poor prognosis based on a physical surrogate, the present study shows that especially relative measures of exercise capacity are associated with malnutrition and inflammation. The Duncan's equation for expected 6MWT distance enables us to analyse exercise capacity in a model controlled for age, gender and BMI. Accordingly, our results indicate that the impact of inflammation and, by extension, MIA, on exercise capacity is different for HD patients of different age, gender and BMI. Based on the equation, we hypothesise that MIA has a greater absolute impact on exercise capacity in younger male subjects with a low BMI compared to female elderly with a high BMI, for a similar relative impairment in exercise capacity.

Although the concept of muscle wasting is included in the definition of PEW, the present study does not strongly confirm this association as similar muscle strength values are noted between groups differing in nutritional status. This finding is in agreement with a study on 330 HD patients reporting that merely 23% of the variance in muscle weakness can be explained by muscle mass [13]. Hence, we posit that the physical screening and rehabilitation of HD patients should focus on functional measures of physical performance rather than only on improving muscle strength.

Ridge regression methods can be used to examine the true global effect of collinear and synergistic determinants (e.g. different measures of nutritional status) on a dependent variable. Borne in mind the high prevalence of frailty in patients with ESKD [51, 52], these true associations between measures of nutritional status and measures of physical performance were lower than expected, albeit nutritional status explained merely 10% of physical performance.

## Limitations

This study has some limitations. First, the cross-sectional nature of this study is a limitation on itself due to the inability to discern temporal relationships and directionality of associations. Second, the original MNA classification was disregarded and an arbitrary cut-off point between the categories of an increased risk for malnutrition and malnutrition was used. Nevertheless, we countered this limitation by focussing the results on the MNA global assessment score. Third, some recommended objective measures of nutritional status could not be obtained due to practical reasons, such as serum albumin and high-sensitive CRP. Hence, conclusions may be biased or missed in the present study. Fourth, the analysis of nutritional status could have benefited from body composition assessments, such as DEXA. Fifth, although various risk of falls assessment tools were included, no history of actual falls was obtained and a few adjustments to the original DFRI were required [24]. Final, as the atherosclerosis component of MIA was not assessed, the involvement of MIA in the association between PEW and physical performance is merely a valid hypothesis.

Nevertheless, this study has several strengths as well. First, a comprehensive examination of physical performance was performed, including a recently developed risk of falls assessment tool tailored to HD patients [24]. Second, objective measures of malnutrition were assessed, which enables us to perform analysis between domains of physical performance and nutritional status. Third, a model that penalizes for multicollinearity and multiple comparisons was used. Fourth and Final, a low threshold for eligibility improved the generalisability of our results to the majority of patients on HD.

## Conclusions and guidelines for further research

In conclusion, this study shows an association between protein-energy wasting and *exercise intolerance* and *gait quality*, to which especially measures of the malnutrition-inflammation-atherosclerosis syndrome and frailty contribute to, respectively. In contrast, malnutrition is

not associated with *muscle strength* in patients on haemodialysis. Future research should aim (1) to further elucidate the relationship between malnutrition and the risk of falls, focussing especially on gait quality, and (2) to assess the effects of interdisciplinary care on nutritional status and physical performance.

The main clinical implication is that nutritionists and physical therapists should collaborate in the rehabilitation of patients on haemodialysis based on the frequently occurring co-existence of exercise intolerance and malnutrition.

## Supporting information

**S1 Table. Dialysis Fall Risk Index.**
(DOCX)

**S2 Table. Patient characteristics according to diabetes.**
(DOCX)

**S3 Table. Detailed association between relevant associations of nutritional measures and domains of physical performance.**
(DOCX)

**S4 Table. Detailed association of relevant associations controlled for age and gender.**
(DOCX)

**S5 Table. Detailed association of relevant associations controlled for age, gender and the Davies comorbidities score.**
(DOCX)

**S6 Table. Detailed association between measures of nutritional status and physical performance as expected for the patients' age and gender.**
(DOCX)

**S7 Table. Detailed association controlled for the Davies comorbidities score.**
(DOCX)

**S1 Dataset. Anonymized dataset.**
(XLSX)

## Acknowledgments

The authors are indebted to Isabel Van Dorpe and Sabine Verhofstede for database management, and to the study nurses Elsie De Man and Kelly Rokegem of the Ghent University Hospital for assistance with the questionnaires.

## Author Contributions

**Conceptualization:** Karsten Vanden Wyngaert, Patrick Calders, Wim Van Biesen, Amaryllis H. Van Craenenbroeck.

**Data curation:** Karsten Vanden Wyngaert, Patrick Calders, Sunny Eloot, Els Holvoet, Wim Van Biesen, Amaryllis H. Van Craenenbroeck.

**Formal analysis:** Karsten Vanden Wyngaert, Patrick Calders, Sunny Eloot, Wim Van Biesen, Amaryllis H. Van Craenenbroeck.

**Investigation:** Patrick Calders, Els Holvoet, Wim Van Biesen, Amaryllis H. Van Craenenbroeck.

**Methodology:** Karsten Vanden Wyngaert, Patrick Calders, Sunny Eloot, Els Holvoet, Wim Van Biesen, Amaryllis H. Van Craenenbroeck.

**Supervision:** Patrick Calders, Sunny Eloot, Wim Van Biesen, Amaryllis H. Van Craenenbroeck.

**Writing – original draft:** Karsten Vanden Wyngaert.

**Writing – review & editing:** Bert Celie, Patrick Calders, Sunny Eloot, Wim Van Biesen, Amaryllis H. Van Craenenbroeck.

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
