## [Decision Letter · Decision Letter 0]

16 Apr 2020

PONE-D-20-01868

Nutritional status and physical performance in haemodialysis patients: a cross-sectional study

PLOS ONE

Dear Dr Van Craenenbroeck,

Thank you for submitting your manuscript to PLOS ONE. After careful consideration, we feel that it has merit but does not fully meet PLOS ONE’s publication criteria as it currently stands. Therefore, we invite you to submit a revised version of the manuscript that addresses the points raised during the review process.

Please considering if the recommendations from the manuscript peer reviewers would strengthen your manuscript. 

We would appreciate receiving your revised manuscript by May 31 2020 11:59PM. To enhance the reproducibility of your results, we recommend that if applicable you deposit your laboratory protocols in protocols.io, where a protocol can be assigned its own identifier (DOI) such that it can be cited independently in the future. For instructions see: http://journals.plos.org/plosone/s/submission-guidelines#loc-laboratory-protocols

We look forward to receiving your revised manuscript.

Kind regards,

Melissa M Markofski

Academic Editor

PLOS ONE

Journal Requirements:

Reviewers' comments:

Reviewer's Responses to Questions

**Comments to the Author**

1. Is the manuscript technically sound, and do the data support the conclusions?

Reviewer #1: Yes

Reviewer #2: Partly

Reviewer #3: Yes

2. Has the statistical analysis been performed appropriately and rigorously? 

Reviewer #1: I Don't Know

Reviewer #2: Yes

Reviewer #3: I Don't Know

3. Have the authors made all data underlying the findings in their manuscript fully available?

Reviewer #1: Yes

Reviewer #2: Yes

Reviewer #3: Yes

4. Is the manuscript presented in an intelligible fashion and written in standard English?

Reviewer #1: Yes

Reviewer #2: Yes

Reviewer #3: Yes

5. Review Comments to the Author

Reviewer #1: Wyngaert et al have submitted a well-written manuscript correlating physical performance with nutritional status in the hemodialysis population.

Line 65: Should the word “relation” actually be “relationship”

Line 76: Unclear what the difference is between high vs low care dialysis units?

Line 103/201/208/229: The use of “drug use” is unclear. Are these illicit drugs? Or medications?

Seems like most of the diabetic patients are in the “bad prognosis, 6MWT < 30m” group. Would this influence your results?

Reviewer #2: The authors performed a cross-sectional analysis comparing markers of physical capacity and nutrition to identify relationships in 113 chronic hemodialysis patients. My concerns about the study are the following:

1. So many cross-sectional analyses are performed that the results are confusing. This is compounded by the fact that the few statistically significant relationships were not adjusted for multiple comparisons.

2. The definition of nutrition (using the mini nutritional status and objective markers) used is flawed. Admittedly, differentiating pure malnutrition from underlying inflammatory processes is difficult, especially in dialysis patients. But it’s not clear to me that C-reactive protein or TIBC or serum total protein are sensitive markers of (under)nutrition. As for BMI, the majority of the study subjects had a BMI in the overweight or obese range. Does this qualify as malnutrition (though it’s actually overnutrition)? Overall the study fails to differentiate inflammation from nutrition.

3. It is unclear how the findings advance in any way our understanding of how malnutrition influences physical health in dialysis patients.

Reviewer #3: Well thought of study. Many limitations especially that one can not have a spacial association but acceptable strengths. It may be interesting to know if there were any difference in diabetics as compared to non diabetics

6. PLOS authors have the option to publish the peer review history of their article (what does this mean?). If published, this will include your full peer review and any attached files.

Reviewer #1: No

Reviewer #2: No

Reviewer #3: Yes: Professor Aasim Ahmad

---

## [Author Response · Author response to Decision Letter 0]

6 Jun 2020

REVIEWER #1

Vanden Wyngaert et al., have submitted a well-written manuscript correlating physical performance with nutritional status in the hemodialysis population.

We thank the reviewer for his/her constructive criticism and interest in our work.

1) Line 65: Should the word “relation” actually be “relationship”

Indeed, the use of the terms relation and relationship have been used interchangeably in the manuscript. To be consistent throughout the manuscript, we changed all relation into relationship.

The following sentences were adapted in the revised version of the manuscript:

Introduction, page 4, lines 66-67: “Nevertheless, to date, the relationship between malnutrition and the different domains of physical performance remains poorly understood [15].”

Conclusions and guidelines for further research, page 19, lines 385-387: “Future research should aim (1) to further elucidate the relationship between malnutrition and the risk of falls, focussing especially on gait quality, and (2) to assess the effects of interdisciplinary care on nutritional status and physical performance.”

2) Line 76: Unclear what the difference is between high vs low care dialysis units?

The indications “low care dialysis” and “high care dialysis” are typically used in Belgium to indicate “satellite unit” and “hospital-based dialysis unit”. We have re-coded this paragraph accordingly. The main intention was to indicate we have included in our cohort a representative mix of patients for the Belgian dialysis landscape. 

The following sentences were adapted in the revised version of the manuscript:

Abstract, page 1, lines 24-26: “This study recruited HD patients between December 2016 and March 2018 from two hospital-based and five satellite dialysis units (registration number on clinicaltrial.gov: NCT03910426).”

Materials and methods, page 5, lines 76-78: “Consecutive patients on maintenance HD in two dialysis centres (including two hospital-based and five satellite dialysis units) were screened for eligibility between December 2016 and March 2018.”

3) Line 103/201/208/229: The use of “drug use” is unclear. Are these illicit drugs? Or medications?

We agree with the reviewer that the use of ‘drug use’ has different meanings in different contexts. Again, to improve the transparency of our manuscript, we adjusted ‘drug use’ by ‘number of prescribed medications’.

The following information was changed in the revised version of the manuscript:

Materials and methods, page 6, lines 104-108: “The screening section addresses food intake, weight loss, mobility, neuropsychological problems, BMI, and health status over the last three months, whereas the in-depth assessment section comprises questions related to the place of residence, number of prescribed medications, skin ulcers, eating and drinking behaviour, the subjective appreciation of nutritional status, and mid-arm and calf circumference.”

Results, page 10, lines 204-206: “Furthermore, these patients were older, more likely to be female, had lower BMI, higher CRP, a higher number of prescribed medications, and scored worse on all other domains of physical performance (Table 1).”

Results, page 10, line 214: “Table 1. Patient characteristics according to prognosis based on a physical surrogate.”

Variable Total

(n=113) Good prognosis (6MWT > 300m,

n=46) Bad prognosis

(6MWT < 300m,

n=67) p value

Number of prescribed medications (n) 13.6 ± 3.7 12.6 ± 3.9 14.3 ± 3.5 0.016

Results, page 12, line 235: “Table 2. Patient characteristics according to nutritional status.”

Variable Normal nutritional status

(n=18) Impaired nutritional status (n=95) p Impaired nutritional status p

(3 groups)

 At risk of malnutrition (n=54) malnutrition (n=41) 

Number of prescribed medications (n) 12.1 ± 4.0 13.9 ± 3.6 0.09 13.5 ± 3.9 14.3 ± 3.1 0.118

4) Seems like most of the diabetic patients are in the “bad prognosis, 6MWT < 300m” group. Would this influence your results?

Indeed, the majority of the patients diagnosed with diabetic nephropathy (i.e. 24 out of 30 patients, 80%) were included in the bad prognosis group. Whether diabetes alone influences the prognosis based on a physical measure is unclear but most likely. Disease burden in type 2 diabetes is high as it is associated with an unhealthy lifestyle, physical inactivity and several comorbidities such as peripheral vascular disease, diabetic neuropathy and musculoskeletal complications such as diabetic foot (Fritschi C., Early declines in physical function among aging adults with type 2 diabetes. Journal of Diabetes and its Complications. 2017). Physical inactivity and comorbid diseases have been associated with impaired physical function in patients with diabetes (Hamasaki H. Daily physical activity and type 2 diabetes: A review. World journal of diabetes. (2016)). Hence, it does not surprise that the proportion of patients with diabetes is higher in the bad prognosis group compared to the good prognosis group based on the 6MWT. 

Based on the reviewers’ suggestion, we performed a between-groups analysis of patients with and without diabetes. The analysis shows a higher age, BMI, CRP concentration and morbidity index in patients with diabetes compared to those without, but no significant differences in measures of physical function were found.

The following information was added to the revised version of the manuscript:

Results, page 10, lines 206-208: “No differences in measures of physical function were found between patients with and without diabetes (Table S2).”

Supplementary files, S2 Table: “S2 Table. Patient characteristics according to diabetes.”

Variable Patients with diabetes (n=52) Patients without diabetes (n=61) p

Age (years) 71 ± 14 63 ± 17 <0.01

BMI (kg/m2) 28 ± 4 24 ± 5 <0.01

CRP (mg/L) 8 ± 7 5 ± 6 0.01

TIBC (µg/dL) 236 ± 77 243 ± 76 0.65

Total protein (g/L) 65 ± 6 64 ± 6 0.48

Dialysis vintage (months) 35 ± 33 36 ± 40 0.86

Number of prescribed medications (n) 14 ± 3 13 ± 4 0.53

Davies comorbidity score (0-7) 2.5 ± 1.1 1.5 ± 1.4 <0.01

Quadriceps force (N) 159 ± 62 182 ± 95 0.13

Quadriceps force (%) 51 ± 19 52 ± 22 0.66

Handgrip force (kg) 27 ± 10 30 ± 12 0.60

Handgrip force (%) 90 ± 28 95 ± 33 0.38

DFRI (/12) 6 ± 3 5 ± 3 0.08

Tinetti (/12) 8 ± 4.2 8 ± 4.7 0.66

FICSIT (/28) 13 ± 7 15 ± 10 0.38

Sit-to-Stand (s) 32 ± 17 27 ± 18 0.12

6MWT (m) 205 ± 164 272 ± 209 0.07

6MWT (%) 36 ± 28 43 ± 31 0.20

Data are reported as mean ± standard deviation; p-values from ANOVA were reported for normal distributed parameters, otherwise they were reported from the Kruskal-Wallis test. Abbreviations: 6MWT, six-minute walking test; BMI, body mass index; DFRI, dialysis fall risk index; CRP, C-reactive protein; TIBC, total iron binding capacity

REVIEWER #2

The authors performed a cross-sectional analysis comparing markers of physical capacity and nutrition to identify relationships in 113 chronic hemodialysis patients. My concerns about the study are the following:

We thank the reviewer for his/her well-taken observations and appreciate the suggestion to substantiate the methodology of this study.

1) So many cross-sectional analyses are performed that the results are confusing. This is compounded by the fact that the few statistically significant relationships were not adjusted for multiple comparisons.

We agree with the reviewer that many cross-sectional studies have been performed and only few have corrected for multiple comparisons. It is indeed important to solely report statistical models that are reliable and correct. Overshooting and overfitting a statistical analysis will result in false positive associations and, consequently, may have a major (adverse) impact on future studies and the standard care of patients. For this reason, we are thrilled to have the opportunity to highlight and explain the statistical background and methods used in this study.

First, the authors chose to use a ridge regression method based on its ability to penalize multicollinearity. Ridge regression as well as Least Absolute Shrinkage and Selection Operator (LASSO) regression models penalize variables with high multicollinearity. A shrinkage estimator (k) is used and produces new estimators that are shrunk closer to the “true” population parameters. For this reason, a ridge regression model is especially good at improving the least-squared estimate when multicollinearity is present. 

Second, ridge regression models can be used to create a parsimonious statistical model when the number of predictor variables exceeds the rule of thumb for the number of observations (15-20 observations per included variable). For each variable added, the estimates will be shrunken more heavily, eventually to the mean, so the estimates will be conservative (Kumamaru H., Dimension reduction and shrinkage methods for high dimensional disease risk scores in historical data. Emerg Themes Epidemiol. 2016 and Pavlou M., How to develop a more accurate risk prediction model when there are few events, BMJ. 2016). In other words, a penalized estimation can be used for small sample sizes and models suffering from multiple comparisons as well as multicollinearity (Muhammad Imdad Ullah., Lecture notes on RR. The R Journal. 2018). The downside of such a model is that, due to the shrinking, the absence of a statistically significant effect does not mean there is no significant effect. It is indeed plausible that associations were missed in our analysis that would have been found in a larger study sample. 

The following sentence was added to the strengths of the present study:

Discussions, pages 18-19, lines 376-377: “Third, a model that penalizes for multicollinearity and multiple comparisons was used.”

2) The definition of nutrition (using the mini nutritional status and objective markers) used is flawed. Admittedly, differentiating pure malnutrition from underlying inflammatory processes is difficult, especially in dialysis patients. But it’s not clear to me that C-reactive protein or TIBC or serum total protein are sensitive markers of (under)nutrition. As for BMI, the majority of the study subjects had a BMI in the overweight or obese range. Does this qualify as malnutrition (though it’s actually overnutrition)? Overall the study fails to differentiate inflammation from nutrition.

We agree with the reviewer that differentiating malnutrition from comorbid conditions is difficult. Patients on dialysis are characterized by various catabolic conditions such as metabolic acidosis, low-grade chronic inflammation, secondary hyperparathyroidism amongst many others. Although these conditions do not necessarily result in pure malnutrition (i.e. poor balance of protein input/output), they will contribute to the clinical presentation of malnutrition substantially. Consequently, the terms Protein-Energy Wasting (PEW) and Malnutrition-Inflammation-Atherosclerosis (MIA-) syndrome were established to describe the multifactorial catabolic environment with the clinical presentation of malnutrition in patients on haemodialysis (Obi Y. Latest consensus and update on protein-energy wasting in chronic kidney disease. Curr Opin Clin Nutr Metab Care. 2015). Also, our research group examined the use of the Mini-Nutritional Assessment scale as a prognostic marker in patients on dialysis and provided data that this screening tool is a relevant measure of nutritional status in this population (Holvoet E., The screening score of Mini Nutritional Assessment is a useful routine screening tool for malnutrition risk in patients on maintenance dialysis. PLOS ONE. 2020). This citation was added to the revised version of the manuscript.

First, we are of the opinion that it is not appropriate – nor clinically relevant - to control for all these different conditions and markers as they coincide in the majority of the malnourished/frail patients with end-stage kidney disease (Kirushnan., Impact of Malnutrition, Inflammation, and Atherosclerosis on the Outcome in Hemodialysis Patients. Journal of nephrology. 2017). Accordingly, the aim of our study was to examine the association between poor protein balance (incl. the related comorbid and contributing factors) and physical function in patients on haemodialysis. Based on the recommendations of Obi et al., we included markers of the following three physiological systems that could influence the association between PEW and physical function: (i) inflammation, measured by CRP and is a marker for MIA-syndrome, (ii) depletion of iron reserves, measured by TIBC and is associated with nutrition-related anaemia, and (iii) protein balance, measured by serum total protein and is associated with the ratio of protein intake/output (Naeeni AE., Assessment of Severity of Malnutrition in Peritoneal Dialysis Patients via Malnutrition: Inflammatory Score. Adv Biomed Res. 2017 and Obi Y., Latest consensus and update on protein-energy wasting in chronic kidney disease. Curr Opin Clin Nutr Metab Care. 2015). By including these measures into a ridge regression, a reliable association between PEW and physical function can be examined. Furthermore, the ridge regression method will penalize the multicollinearity between these markers, ensuring that our model would not overestimate the true association.

Second, we agree with the reviewer that BMI is a poor indicator of nutritional status. Although BMI in the overweight-obese ranges is associated with increased cardiovascular risk and decreased survival in the general population, the opposite can been observed in patients with end-stage kidney disease (Calabia J., Does the obesity survival paradox of dialysis patients differ with age?. Blood Purif. 2015). Higher BMI has been paradoxically associated with better survival in patients with end-stage kidney disease. Recent data indicate that both higher skeletal muscle mass as well as increased total body fat are protective, albeit the data on increased visceral (intra-abdominal) fat is inconclusive (Ghorbani A., The prevalence of malnutrition in hemodialysis patients. J Renal Inj Prev. 2020 and Obi Y., Latest consensus and update on protein-energy wasting in chronic kidney disease. Curr Opin Clin Nutr Metab Care. 2015). Possible causes of this paradox include e PEW, MIA, inflammation, hemodynamic stability, sequestration of uremic toxins in adipose tissue amongst many others. The obesity paradox dovetails with the aim of this study to focus on PEW rather than only on pure nutritional status (Naderi N., Obesity Paradox in Advanced Kidney Disease: From Bedside to the Bench. Prog Cardiovasc Dis. 2018 and Park J., Obesity paradox in end-stage kidney disease patients. Prog Cardiovasc Dis. 2014). Last, we believe the reviewer is indeed correct to state that at least part of patients with high BMI might be malnourished, especially from the protein/anabolic perspective, as in patients on dialysis, there is a tendency for excess fat mass and decreased lean tissue mass. In this regard, some patients with high BMI can be considered “malnourished” (Van Biesen W., A multicentric, international matched pair analysis of body composition in peritoneal dialysis versus haemodialysis patients. Nephrol Dial Transplant. 2013).

3) It is unclear how the findings advance in any way our understanding of how malnutrition influences physical health in dialysis patients.

Impairments in physical health and physical function are common in patients on dialysis and these impairments have a multifactorial aetiology. Decreased physical functioning is closely related to objective and subjective health-related quality of life. Exercise training programmes have shown to improve physical function in patients on dialysis, but levels remain considerably below the recommended/expected levels of physical function and physical activity. That is why unravelling of the role of the different players is important as a better understanding of the underlying mechanisms might lead to better exercise training outcomes in patients on haemodialysis. This study advances our understanding of malnutrition in two ways: 1/ by differentiating between aspects of physical function and identifying those aspects that are or are not associated with PEW, and 2/ by examining the true association between PEW (instead of pure malnutrition) and physical function.

1/ Poor nutritional status has been associated with muscle strength and the risk of falls in patients on haemodialysis, albeit the literature on the strength of these associations is inconclusive. To our knowledge, this study is the first to differentiate between different aspects of physical function (i.e. functional vs. absolute measures of physical function and measures of endurance vs. strength vs. balance). The results of our study indicate that PEW affect endurance capacity and gait quality in patients on haemodialysis but not muscle strength. 

2/ Various cross-sectional studies have examined the relationship between malnutrition and physical function (mainly muscle strength). However, to our knowledge, a cross-sectional relationship between PEW (defined as malnutrition together with a marker indicating a catabolic environment and analysed by a ridge regression model to correct for multicollinearity) and different aspects of physical function has not yet been examined. The term PEW is a composite of malnutrition and pro-malnutrition markers as proposed by Rodrigues et al., (Juliana Rodrigues., Nutritional assessment of elderly patients on dialysis: pitfalls and potentials for practice, Nephrology Dialysis Transplantation, 2017). Using a ridge regression to incorporate the weight of the different markers, the final marker gives a representation of PEW in general. This methodological approach allows to avoid the problem of multi-collinearity and multiple testing when different markers are each introduced individually in the model and is thus a more-sound approach from the statistical/methodological approach. Furthermore, our results suggest that the MIA-syndrome is an important contributor to exercise intolerance in patients on haemodialysis.

REVIEWER #3

Well thought of study. Many limitations especially that one can not have a spacial association but acceptable strengths. It may be interesting to know if there were any difference in diabetics as compared to non diabetics

We thank the reviewer for this pertinent remark, which is in line with the last comment of reviewer 1. Indeed, diabetes is a syndrome which has an impact on both nutritional as well as physical status. Whether diabetes alone influences the prognosis based on a physical measure is unclear but most likely. Disease burden in type 2 diabetes is high as it is associated with an unhealthy lifestyle, physical inactivity and several comorbidities such as peripheral vascular disease, diabetic neuropathy and musculoskeletal complications such as diabetic foot (Fritschi C., Early declines in physical function among aging adults with type 2 diabetes. Journal of Diabetes and its Complications. 2017). Physical inactivity and comorbid diseases have been associated with impaired physical function in patients with diabetes (Hamasaki H. Daily physical activity and type 2 diabetes: A review. World journal of diabetes. (2016)). Hence, it does not surprise that the proportion of patients with diabetes is higher in the bad prognosis group compared to the good prognosis group based on the 6MWT. 

Based on the reviewers’ suggestion, we performed a between-groups analysis of patients with and without diabetes. The analysis shows a higher age, BMI, CRP concentration and morbidity index in patients with diabetes compared to those without, but no significant differences in measures of physical function were found.

The following information was added to the revised version of the manuscript:

Results, page 10, lines 206-208: “No differences in measures of physical function were found between patients with and without diabetes (Table S2).”

Supplementary files, S2 Table: “S2 Table. Patient characteristics according to diabetes.”

Variable Patients with diabetes (n=52) Patients without diabetes (n=61) p

Age (years) 71 ± 14 63 ± 17 <0.01

BMI (kg/m2) 28 ± 4 24 ± 5 <0.01

CRP (mg/L) 8 ± 7 5 ± 6 0.01

TIBC (µg/dL) 236 ± 77 243 ± 76 0.65

Total protein (g/L) 65 ± 6 64 ± 6 0.48

Dialysis vintage (months) 35 ± 33 36 ± 40 0.86

Number of prescribed medications (n) 14 ± 3 13 ± 4 0.53

Davies comorbidity score (0-7) 2.5 ± 1.1 1.5 ± 1.4 <0.01

Quadriceps force (N) 159 ± 62 182 ± 95 0.13

Quadriceps force (%) 51 ± 19 52 ± 22 0.66

Handgrip force (kg) 27 ± 10 30 ± 12 0.60

Handgrip force (%) 90 ± 28 95 ± 33 0.38

DFRI (/12) 6 ± 3 5 ± 3 0.08

Tinetti (/12) 8 ± 4.2 8 ± 4.7 0.66

FICSIT (/28) 13 ± 7 15 ± 10 0.38

Sit-to-Stand (s) 32 ± 17 27 ± 18 0.12

6MWT (m) 205 ± 164 272 ± 209 0.07

6MWT (%) 36 ± 28 43 ± 31 0.20

Data are reported as mean ± standard deviation; p-values from ANOVA were reported for normal distributed parameters, otherwise they were reported from the Kruskal-Wallis test. Abbreviations: 6MWT, six-minute walking test; BMI, body mass index; DFRI, dialysis fall risk index; CRP, C-reactive protein; TIBC, total iron binding capacity

---

## [Decision Letter · Decision Letter 1]

17 Jun 2020

PONE-D-20-01868R1

Nutritional status and physical performance in haemodialysis patients: a cross-sectional study

PLOS ONE

Dear Dr. Van Craenenbroeck,

Thank you for submitting your manuscript to PLOS ONE. After careful consideration, we feel that it has merit but does not fully meet PLOS ONE’s publication criteria as it currently stands. Therefore, we invite you to submit a revised version of the manuscript that addresses the points raised during the review process.

There are two comments from reviewers for consideration:

1) consider changing the title (not required, but the suggested title may be more accurate);

2) carefully read through the manuscript and check that it is in final form. PLOS ONE does not have a copy editor review the manuscript prior to publication, so the authors must ensure that it meets point five of the PLOS ONE publication criteria. 

We look forward to receiving your revised manuscript.

Kind regards,

Melissa M Markofski

Academic Editor

PLOS ONE

Reviewers' comments:

Reviewer's Responses to Questions

**Comments to the Author**

1. If the authors have adequately addressed your comments raised in a previous round of review and you feel that this manuscript is now acceptable for publication, you may indicate that here to bypass the “Comments to the Author” section, enter your conflict of interest statement in the “Confidential to Editor” section, and submit your "Accept" recommendation.

Reviewer #1: All comments have been addressed

Reviewer #2: (No Response)

Reviewer #3: All comments have been addressed

2. Is the manuscript technically sound, and do the data support the conclusions?

Reviewer #1: Yes

Reviewer #2: Yes

Reviewer #3: Yes

3. Has the statistical analysis been performed appropriately and rigorously? 

Reviewer #1: I Don't Know

Reviewer #2: Yes

Reviewer #3: I Don't Know

4. Have the authors made all data underlying the findings in their manuscript fully available?

Reviewer #1: Yes

Reviewer #2: Yes

Reviewer #3: Yes

5. Is the manuscript presented in an intelligible fashion and written in standard English?

Reviewer #1: Yes

Reviewer #2: Yes

Reviewer #3: No

6. Review Comments to the Author

Reviewer #1: (No Response)

Reviewer #2: I would suggest that you change the title of the manuscript to: Markers of Protein-Energy Wasting and Physical Performance in Haemodialysis Patients: A Cross-Sectional Study

This is more accurate than using "Nutritional Status" in the title since strictly speaking you are not measuring nutritional status

Reviewer #3: (No Response)

7. PLOS authors have the option to publish the peer review history of their article (what does this mean?). If published, this will include your full peer review and any attached files.

Reviewer #1: No

Reviewer #2: No

Reviewer #3: Yes: Aasim Ahmad

---

## [Author Response · Author response to Decision Letter 1]

23 Jun 2020

REVIEWER #1 and #3

We thank the reviewers for their feedback.

REVIEWER #2

1) I would suggest that you change the title of the manuscript to: Markers of Protein-Energy Wasting and Physical Performance in Haemodialysis Patients: A Cross-Sectional Study. This is more accurate than using "Nutritional Status" in the title since strictly speaking you are not measuring nutritional status.

We agree with the reviewer. Changing “nutritional status” to “markers of protein-energy wasting” is more in line with the true message of our manuscript.

The following adjustments were made in the title of the revised version of the manuscript:

Title page, page 1, lines 1-2: “Markers of protein-energy wasting and physical performance in haemodialysis patients: a cross-sectional study”

---

## [Editor Report · Decision Letter 2]

15 Jul 2020

Markers of protein-energy wasting and physical performance in haemodialysis patients: a cross-sectional study

PONE-D-20-01868R2

Dear Dr. Van Craenenbroeck,

We’re pleased to inform you that your manuscript has been judged scientifically suitable for publication and will be formally accepted for publication once it meets all outstanding technical requirements.

Kind regards,

Melissa M Markofski

Academic Editor

PLOS ONE
---

## [Editor Report · Acceptance letter]

17 Jul 2020

PONE-D-20-01868R2 

Markers of protein-energy wasting and physical performance in haemodialysis patients: a cross-sectional study 

Dear Dr. Van Craenenbroeck:

I'm pleased to inform you that your manuscript has been deemed suitable for publication in PLOS ONE. Congratulations! Your manuscript is now with our production department. 

Kind regards, 

on behalf of

Dr. Melissa M Markofski 

Academic Editor

PLOS ONE